# Genomic Insights into High-Altitude Adaptation: A Comparative Analysis of *Roscoea alpina* and *R. purpurea* in the Himalayas

**DOI:** 10.3390/ijms25042265

**Published:** 2024-02-14

**Authors:** Ya-Li Wang, Li Li, Babu Ram Paudel, Jian-Li Zhao

**Affiliations:** 1Ministry of Education Key Laboratory for Transboundary Ecosecurity of Southwest China, Yunnan Key Laboratory of Plant Reproductive Adaptation and Evolutionary Ecology and Centre for Invasion Biology, Institute of Biodiversity, School of Ecology and Environmental Science, Yunnan University, Kunming 650504, China; wangyali@ynu.edu.cn (Y.-L.W.); lili0426@ynu.edu.cn (L.L.); 2Research Centre for Applied Science and Technology, Tribhuvan University, Kirtipur 44613, Nepal

**Keywords:** alpine ginger, genomic divergence, environmental stress, high-altitude adaptation, the Himalayas, *Roscoea*

## Abstract

Environmental stress at high altitudes drives the development of distinct adaptive mechanisms in plants. However, studies exploring the genetic adaptive mechanisms of high-altitude plant species are scarce. In the present study, we explored the high-altitude adaptive mechanisms of plants in the Himalayas through whole-genome resequencing. We studied two widespread members of the Himalayan endemic alpine genus *Roscoea* (Zingiberaceae): *R. alpina* (a selfing species) and *R. purpurea* (an outcrossing species). These species are distributed widely in the Himalayas with distinct non-overlapping altitude distributions; *R. alpina* is distributed at higher elevations, and *R. purpurea* occurs at lower elevations. Compared to *R. purpurea*, *R. alpina* exhibited higher levels of linkage disequilibrium, Tajima’s *D*, and inbreeding coefficient, as well as lower recombination rates and genetic diversity. Approximately 96.3% of the genes in the reference genome underwent significant genetic divergence (*F*_ST_ ≥ 0.25). We reported 58 completely divergent genes (*F*_ST_ = 1), of which only 17 genes were annotated with specific functions. The functions of these genes were primarily related to adapting to the specific characteristics of high-altitude environments. Our findings provide novel insights into how evolutionary innovations promote the adaptation of mountain alpine species to high altitudes and harsh habitats.

## 1. Introduction

High-altitude environments are characterized by high ultraviolet radiation, low temperature, hypoxia, and reduced incidence of pathogens [1,2]. To survive and be able to inhabit such harsh environments, local species have evolved effective strategies for the adaptation of genes to specific morphological and physiological traits [3,4,5,6,7]. Plant populations across altitude gradients exhibit genetic differentiation and local adaptation to specific environmental conditions [8,9]. High-altitude plants often exhibit genetic adaptations for cold tolerance to withstand freezing temperatures and frost [10]. They have genetic adaptations to cope with high light intensity and UV radiation [11]. They often possess genetic adaptations for efficient photosynthesis under low-CO_2_ conditions [12]. They exhibit genetic adaptations to cope with hypoxic conditions, enabling them to maintain energy production and metabolic homeostasis under hypoxic stress [13]. High-altitude plants face water scarcity and drought stress, especially in arid or alpine environments. They possess genetic adaptations for water conservation and drought resistance [14]. For instance, corn cultivated in high-altitude areas frequently accumulates flavonoids within its leaves and filaments to mitigate the effects of high UV-B exposure [15,16]. Genetic adaptations in plants to high-altitude environments are diverse and complex, enabling them to thrive in harsh conditions. Under extreme conditions, such as those in high-altitude regions, natural selection can drive quick alterations in allele frequencies to optimally enhance adaptability [5]. In recent years, there has been a growing interest in assessing genomic variations in natural populations by identifying adaptive loci to understand how organisms adapt to various habitats [17,18]. The development of high-throughput sequencing technology has greatly accelerated genomic research and identification of key genes and promoted the adaptive evolution and ecological research of non-model organisms [19,20]. This technology is also beneficial for further exploring the adaptation of non-model plants to high elevation [8,9,21,22].

The Himalayas, located on the southern margin of the Tibetan Plateau, have an elevation gradient of over 8000 m from south to north within a narrow latitude range [23,24] and are a biodiversity hotspot. Records of endemic species in the Himalayas and recent findings suggest that in situ speciation, especially divergence along the elevational gradient, plays a significant role in the region’s high biodiversity [25,26,27]. High-elevation species in the Himalayas face more rigorous environmental challenges than those experienced by lower-elevation species. Thus, exploring the genetic adaptation and divergence of high-altitude plants can provide important insights into their survival mechanisms in the harsh environments of the high Himalayas.

*Roscoea*, the only alpine genus in the pantropical family Zingiberaceae, is distributed at elevations ranging from approximately 1200 to 4800 m. *Roscoea* species have been categorized into two distinct groups: the Himalayas clade and the Hengduan Mountains clade [28,29,30,31]. Species in the Himalayas clade generally exhibit a distribution pattern along the altitude gradient (approximately 1200–4500 m). This distinct altitude divergence among the Himalayas *Roscoea* species is associated with the rapid uplift of the Himalayas and climate change [27]. *R. alpina* and *R. purpurea* are two widely distributed species along the Himalayas from west to east without an overlapping distribution [27,29]. Autonomous selfing is the predominant reproductive mode for *R. alpina* [32], whereas outcrossing is the reproductive strategy of *R. purpurea* [32,33]. Among the species in the genus *Roscoea*, *R. alpina* has the highest elevation and mainly occurs in alpine meadows, whereas *R. purpurea* is mainly distributed at lower elevations and found growing under trees [27,29]. Previous phylogenetic reconstruction based on restriction association site DNA (RAD) revealed that *R. alpina* diverged from the ancestor of *R. purpurea* approximately 14 million years ago (Ma) [27]. Thus, these two related species provide good models for elucidating the adaptive mechanisms of plants to high-altitude environments in the Himalayas. The aim of the present study was to investigate the adaptive mechanisms of *R. alpina* to high-altitude environments in the Himalayas by using whole-genome resequencing.

## 2. Results

### 2.1. Genomic Feature Investigation

To obtain insight into natural selection patterns and historical aspects of population growth, genome-wide patterns of linkage disequilibrium (LD) were determined. The average *r* square (*r*^2^) in the LD tended to decrease with increasing distances between pairwise single-nucleotide polymorphisms (SNPs), with a rapidly declining trend observed over the first 0.2 kb. The LD decay of *R. alpina* and *R. purpurea* revealed a similar declining trend (Figure 1a). However, a significant difference in *r*^2^ was observed over the same distance between the two species (*p* < 0.05). The highest *r*^2^ of *R. alpina* (approximately 0.8) was in the short-distance bin <300 bp, and it declined to the lowest *r*^2^ (approximately 0.45) at the longest-distance of approximately 0.7 kb. The highest *r*^2^ of *R. purpurea* (approximately 0.56) was in the short-distance bin <300 bp, and it declined to the lowest *r*^2^ (approximately 0.26) in the longest-distance bin of approximately 0.65 kb. The *t*-test results of the mean *r*^2^ in each 1 kb bin showed a significant difference between the two species (Figure 1a). The population recombination rates corresponding to LD decay indicated a higher recombination rate in *R. purpurea* on 12 chromosomes than in *R. alpina* (Figure 1b).

Tajima’s *D*, π, *F*_IS_, and genome heterozygosity (*H*_E_) were used as indicators of genetic diversity; they were compared between *R. purpurea* and *R. alpina*. The results indicate that *R. alpina* has lower genetic variation than *R. purpurea*. Tajima’s *D* and *F*_IS_ of *R. alpina* were significantly higher than those of *R. purpurea* (Figure 2a,b), whereas π and *H*_E_ of *R. alpina* were significantly lower than those of *R. alpina* (Figure 2c,d). 

To eliminate the effects of sampling size on comparisons of the parameters between the two species, random sampling analysis was conducted. Although Tajima’s *D* and π varied with sampling size, Tajima’s *D* of all random sampling in *R. alpina* was significantly higher than that of all random sampling in *R. purpurea* (Appendix A), and π of all random sampling in *R. alpina* was significantly lower than π of all random sampling in *R. purpurea* (Appendix A). In addition, the variation tended to be stable when the sampling size was seven, indicating that seven individuals had sufficient SNPs for estimating Tajima’s *D* and π in the present study (Appendix A). 

### 2.2. Difference in Demographic History

Different individuals of each species had a similar demographic history, but it was significantly different between *R. purpurea* and *R. alpina* (Figure 3). During the 1.6–1.0 Ma period, the effective population size of *R. alpina* was greater than that of *R. purpurea*. The effective population size of *R. purpurea* began to decline around 1.5 Ma, and the decline lasted until approximately 50,000 years ago. The effective population size of *R. alpina* started to decline sharply at approximately 1.0 Ma and declined until approximately 15,000 years ago. Approximately 10,000 years ago, the effective population size of *R. purpurea* was twice that of *R. alpina*.

### 2.3. Candidate Genes Associated with High-Altitude Adaptation

Generally, the functions of divergent genes selected by *F*_ST_ are potentially related to adaptability [34,35,36]. We used *F*_ST_ to investigate the genes related to high-altitude adaptation in *R. alpina*. The random sampling size for *F*_ST_ was analyzed to eliminate the effects of sampling size on *F*_ST_ estimation and the subsequent search for potentially adaptive genes. The random sampling showed similar results; most windows and genes of all random sampling were highly divergent. *R. purpurea* sample size did not influence the *F*_ST_ distribution landscape (Appendix A). A total of 17,108 highly genetically divergent windows (*F*_ST_ > 0.25) were identified (including 96.3% of the genes in the reference genome) (Figure 4). 

The genes with *F*_ST_ = 1 have the highest degree of divergence in the genome. Such a high degree of divergence suggests that the genes may play an important role in environmental adaptation. To eliminate the influence of sample size bias and extract more reliable environmental adaptive genes, genes in *F*_ST_ = 1 windows were extracted by all random sampling and non-random sampling strategies. There were 76 common windows across all sampling strategies (Appendix A), and 58 genes were annotated in the windows (Appendix A). Among the 58 genes, 17 genes were annotated with specific functions. Most of the gene functions were associated with responses to environmental stress, DNA repair, and photosynthesis (Table 1). However, 41 of the 58 genes’ functions were unknown (Appendix A), and we speculated that their functions may be related to high-altitude adaptation.

## 3. Discussion

### 3.1. Genomic Features for Adaptive Evolution in the High Himalayas

We speculated that different reproductive strategies and selection pressures may lead to differences in LD decay and population recombination rate. Self-fertilization is an adaptive strategy for plants in harsh environments, such as those where pollinators are absent [36,53,54]. Self-fertilization provides reproductive assurance under low levels of insect diversity in alpine ecosystems [55,56,57,58]. Autonomous selfing in *R. alpina* has been proposed as an evolutionary strategy for reproductive success in the alpine zone of the Himalayas [32]. Inbreeding and selfing have been observed to increase the correlation between alleles at different loci, contributing to increased LD and decreased recombination rate [36,54,59,60,61]. Contrasting LD and recombination rates between these two species are likely associated with their different mating systems, namely autonomous selfing in *R. alpina* [62] and outcrossing in *R. purpurea* [32,33]. Similar results were observed in maize and *Arabidopsis* [63]. In some cases, selection can increase the LD [64]. When interacting loci are closely linked or selection is strong, the recombination rate is likely to decrease [65,66]. The lower recombination rate and higher LD levels in *R. alpina* suggest that it may have undergone strong natural selection.

Long-term selfing may decrease genetic diversity and enhance linkage effects in the genome [61,67], consistent with our findings of lower π and *H*_E_ and higher *F*_IS_ in *R. alpina* than in *R. purpurea*. High LD is predicted to decrease the polymorphism of the linked loci, which may eventually lead to a significant decrease in the genetic diversity of *R. alpina*. Positive Tajima’s *D* values were observed in both *R. alpina* and *R. purpurea*, which, combined with the demographic results (Figure 4), suggest that the two species have undergone genetic bottlenecks [68,69]. However, higher Tajima’s *D* values suggest that *R. alpina* has undergone a stronger genetic bottleneck in comparison with *R. purpurea*. A strong genetic bottleneck has also been observed in other alpine plants [70,71]. Therefore, the lower genetic diversity of *R alpina* could be the consequence of adaptive evolution to the higher elevation in the Himalayas [71,72]. 

### 3.2. Difference in Demographic History within the Himalayas

The difference in demographic history suggests that *R. alpina* and *R. purpurea* in the Himalayas may respond differently to climate change. Under the influence of changing climate, the habitable area for the species shifts toward mountain tops and thus becomes narrower, with the habitats becoming harsher [73,74,75]. Consequently, colonization of higher mountain elevations should result in stronger genetic bottlenecks/drift and a sharp decrease in effective population size, as indicated by our findings in *R. alpina*. When the effective population size of R. alpina increased to the maximum from ~1.6–1.2 Ma, temperature variation between glaciations and interglaciations was relatively stable, not reaching full glacial values [76]. However, the maximum increase in the effective population size of *R. purpurea* occurred from ~1.8–1.5 Ma, and its population size began to decline at the onset of the Ice Age (~1.5 Ma). The decline in the effective population size of *R. alpina* was delayed by about 0.5 Ma compared to that of *R. purpurea*. We speculated that *R. alpina* may have long-term adaptations to low-temperature environments at higher altitudes, which is why its survival would have been largely unaffected until the temperature dropped to full glacial values.

Recombination rate is positively correlated with effective population size and genetic diversity [77,78,79]. The lower genetic diversity (Figure 2c) and recombination rate (Figure 1b) related to the selfing characteristics of *R. alpina* could have lagged behind *R. purpurea* in restoring the effective population size. The higher genetic diversity (Figure 2c) and recombination rate (Figure 1b) of *R. purpurea* could have improved its ability to restore the effective population size because, after genetic bottlenecks, outcrossing species with higher recombination rates and genetic diversity possess a greater ability to increase their effective population size [77,80,81].

### 3.3. Candidate Genes Associated with High-Altitude Adaptation

At high altitudes in the Himalayas, the most severe environmental stresses include extreme cold, low oxygen levels, high UV radiation, pathogens, and other biotic and abiotic stressors [82,83]. Plants that inhabit the Himalayas have evolved in their morphological structure, physiology, and metabolism to adapt to the extreme ecological conditions of this region. Their evolutionary genetic changes often adhere to certain patterns, evident in factors such as cold tolerance, efficiency of photosynthesis, hypoxia tolerance, antioxidant defense mechanisms, stress response, and drought resistance. We found several completely divergent genes that were likely associated with alpine adaptations in *R. alpina* (Table 1). *AAEs* and *VAR3* genes can help species adjust their secondary metabolite production to cope with harsh environments. The *RFS2*, *RLK*, and *PER65* genes were also related to stress responses, and *SHAT1-5*, *BRs*, *REL2*, *E2*, *CALS9*, *StEXPA3*, *RPN8a*, and *MEE40* play important roles in the response to biotic and abiotic stress at high altitudes. Based on our observations, budding and flowering times differed between *R. alpina* and *R. purpurea*. The *CK2* gene may regulate the circadian clock to adjust to the unpredictable climate between higher and lower altitudes in the Himalayas. The *FAR1* and *POD* genes can improve the photosynthetic rate by regulating the synthesis of sucrose and starch. The *AtLPP1* gene can repair the DNA damage caused by high UV and solar radiation. Notably, among these key genes, *E2*, *RLK*, and *FAR1* have been proposed to facilitate the adaptation of alpine plants to high-altitude environments through convergent evolution [9,84]. These genes could have facilitated *R. alpina* adaptation to higher elevation, with extensive distribution along the Himalayas. 

## 4. Materials and Methods

### 4.1. Resequencing and Variant Discovery

Seven individuals of *R. alpina* and thirteen individuals of *R. purpurea* were collected from wild populations in the Himalayas (Figure 5, Appendix A). To obtain a sufficient number of SNPs, a whole-genome resequencing depth greater than 30× was adopted for SNP extraction. For genome sequencing, at least 5 μg of genomic DNA was extracted from fresh leaves by using the cetyltrimethylammonium bromide (CTAB) method [85]. DNA libraries were constructed and barcoded by using the DNA Library Prep Reference Guide (Illumina, Inc., San Diego, CA, USA). After sequencing on the Illumina Hiseq X Ten platform, 150 paired-end whole-genome sequencing reads with an insert size of 350 bp were obtained. The average sequencing depth was >30×, and 1072 Gb of raw sequencing data were obtained, with an average of 53.60 Gb per sample (Appendix A).

FastQC v.0.11.9 was used to assess the quality of raw data (https://www.bioinformatics.babraham.ac.uk/projects/fastqc/ (accessed on 8 January 2019)). Subsequently, Trimmomatic v.0.36 [86] was used to filter the sequences. First, the first 15 bp potential adaptor sequences evaluated by FastQC were removed. Second, low-quality paired reads with more than 10% unrecognized bases were eliminated. Third, low-quality bases with Phred quality scores <30 were trimmed. After filtering, the reads were aligned to the reference genome of *Roscoea schneideriana* (unpublished) by using BWA-MEM v.0.7.17 [87] with the default parameters. SAMtools v.1.12 (https://sourceforge.net/projects/samtools/ (accessed on 17 March 2021)) [88] was used to convert the mapping results into the BAM format and filter the unaligned and non-unique aligned reads. Duplicated reads were marked and filtered by using Picard v.2.1.1 (picard.sourceforge.net (accessed on 4 March 2016)). After mapping, the reads were realigned by using the Genome Analysis Toolkit (GATK) v.3.8 (https://hub.docker.com/r/broadinstitute/gatk3/tags/ (accessed on 28 July 2017)) [89] in two steps. In the first step, the “RealignerTargetCreator” package was used to identify regions where realignment was required. In the second step, the “IndelRealigner” package was used to realign the regions found in the first step to produce a realigned BAM file for each sample.

Variation detection with the realigned BAM file followed the best-practice workflow recommended by GATK [89]. Briefly, the variants were called for each individual by using the GATK HaplotypeCaller. A joint genotyping step for a comprehensive variation union was performed by using the gVCF files. In the hard filtering step, the SNP filter expression was set as “QD < 2.0 || MQ < 40.0 || FS > 60.0 || SOR > 3.0 || MQRankSum < −12.5 || ReadPosRankSum < −8.0 || QUAL < 30”. PLINK v.1.90 (https://www.cog-genomics.org/plink/ (accessed on 15 May 2014)) was used to further filter the SNPs with the parameters “--geno 0.1 --maf 0.01”. Finally, 1,776,773 filtered SNPs were obtained for subsequent analysis.

### 4.2. Genomic Feature Investigation

LD was calculated based on the correlation coefficient (*r*^2^) statistics for genome-wide filtered SNPs by using PopLDdecay v.3.29 (https://github.com/BGI-shenzhen/PopLDdecay (accessed on 11 September 2018)) with the default parameters. An LD decay plot was made by using Perl script Plot_MultiPop.pl, with the parameters set as “-bin1 1 -bin2 10 -maxX 0.8”. To conduct statistical tests for LD differences, geno-r2 of LD statistics in VCFtools v.0.1.13 (https://vcftools.sourceforge.net (accessed on 3 August 2015)) [90] was used to calculate the average *r*^2^ of each 1 kb bin. Subsequently, a paired *t*-test for *r*^2^ between *R. alpina* and *R. purpurea* was performed by using the R program (https://www.R-project.org/ (accessed on 21 April 2023)).

To estimate the population recombination rate (*ρ*), Beagle v.5.2 (https://faculty.washington.edu/browning/beagle/b5_2.html (accessed on 28 January 2021)) [91] was used to phase the filtered SNPs, and the phased data were then input into the FastEPRR_VCF_step1 function in FastEPRR v.2.0 (https://www.picb.ac.cn/evolgen/softwares/download/FastEPRR/FastEPRR2.0/ (accessed on 10 January 2021)) [92] to scan the sequences and store the required information in files for each 50 kb window with the parameters winLength = 50,000 and winDXThreshold = 10. Subsequently, FastEPRR_VCF_step2 was used to estimate the recombination rate for each window. Finally, FastEPRR_VCF_step3 was used to merge the files generated by step 2 for each chromosome.

Tajima’s *D* and π of *R. alpina* and *R. purpurea* were computed by using VCFtools. *F*_IS_ of the two species was calculated by using PLINK. KmerGenie v.1.7048 (http://kmergenie.bx.psu.edu (accessed on 14 March 2018)) [93] was used to estimate the optimal k-mer length for the de novo genome assembly. GenomeScope v.1.0 (https://github.com/schatzlab/genomescope/ (accessed on 15 January 2017)) [94] was used to estimate genome heterozygosity (*H*_E_). The *t*-test for the four genetic diversity parameters between *R. alpina* and *R. purpurea* was performed using the R program.

Although our sample size is small, other studies have shown that a small sample size (as small as *n* = 4–6) with a sufficient number of SNPs (at least 3000 SNPs) can estimate parameters of population genomics accurately, including genetic diversity [95,96]. In addition, to test whether our genetic diversity results could be affected by sample size, random sampling strategies were adopted for the calculation of genetic diversity. Four, five, six, and seven individuals of each species were selected randomly to calculate genetic diversity. We adopted ten times random sampling for each sample size. The Tajima’s *D* and π of each random sampling were calculated for both species. The parameters between species under different sample sizes were compared to test the impact of sample size on the two values. 

### 4.3. Demographic Inferences

Consensus sequences were used to estimate demographic history. Individual paired-end whole-genome sequencing reads were mapped to reference genomes by using the subscripts mpileup and the call format of BCFtools v.0.1.17 (https://www.htslib.org/doc/1.0/bcftools.html (accessed on 7 July 2017)), and vcf files of consensus sequences were generated. Subsequently, vcfutils.pl was used to convert the consensus sequence vcf files into consensus FASTQ sequence files, and then the FASTQ files were converted to consensus FASTA sequences by using SeqTK v.1.3 (https://github.com/lh3/seqtk/ (accessed on 18 June 2018)). The PSMC v.0.6.5 (https://github.com/lh3/psmc/ (accessed on 30 April 2015)) model was used to examine the demographic history of each individual [97]. It was run for a total of 25 iterations with the parameters “-t15 -r5 -p ‘4 + 25 × 2 + 4 + 6’”, a generation time of two years, and a mutation rate of 6.36 × 10^−9^ per site per generation.

### 4.4. Identification of Candidate Genes Associated with High-Altitude Adaptation

Several researchers have used *F*_ST_ = 1 to search adaptive candidate genes between related species [98,99,100]. We used the same strategy to obtain the genes potentially related to high-altitude adaptation. The *F*_ST_ values across the *Roscoea* genome (window size = 50,000 bp) were used to identify the candidate genomic regions. The genomic windows with *F*_ST_ = 1 were treated as candidate high-altitude adaptation windows. 

The sample size of *R. purpurea* is nearly twice that of *R. alpina*. The imbalance in sample size may skew the *F*_ST_ results. To exclude the bias and evaluate the effect of sample size on *F*_ST_, seven, eight, nine, ten, eleven, and twelve individuals of *R. purpurea* were selected randomly for estimating *F*_ST_ under a stable sample size of *R. alpina* of seven. The number of times adopted for random sampling of each sample size was ten. Subsequently, the windows with *F*_ST_ = 1 were extracted. The windows shared by 10 random sampling times under the same sample size and all random sampling were extracted. Genes were extracted if the regions overlapped with the final extracted windows. The gene sequences were annotated to the NR database by using BLAST v. 2.11s. Homologous genes in the NR database were retained. The functions of the homologous genes were found in the literature. Finally, the genes with functional annotations were identified as candidate genes.

## 5. Conclusions

In the present study, we compared the genetic differentiation between the high-altitude *R. alpina* and low-altitude *R. purpurea* in the Himalayas and explored whether their genetic variation is associated with adaptation to high-altitude environments. Our results suggested that perhaps under the influence of selective pressure and autonomous selfing characteristics, the high-altitude *R. alpina* exhibited a high level of LD, low population recombination rate, delayed decrease in effective population size, and smaller effective population size. Additionally, it had high values of Tajima’s *D* and *F*_IS_ while displaying low π and *H*_E_. Founder effect and lower recombination rate (higher LD) correlation with lower genetic diversity of *R. alpina* and inbreeding may enhance the LD. Furthermore, selection possibly reduced the nucleotide diversity of genomic loci and reduced the diversity of its linkage loci. Due to its long-term survival and propagation in the Himalayas, this species has evolved hereditary genes that protect it from extremely harsh high-altitude environments. Moreover, we identified numerous high-altitude adaptation-related genes that diverged during the adaptation to high-altitude environments. These completely divergent genes may be the evolutionary innovations of *R. alpina* driven by harsh environmental stresses. 

## Figures and Tables

**Figure 1 ijms-25-02265-f001:**
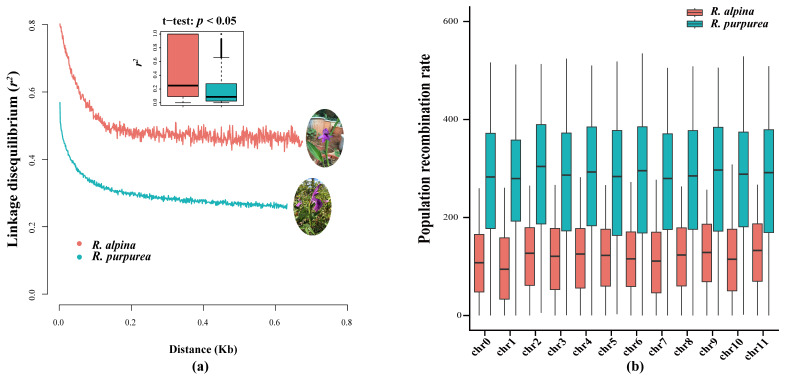
Linkage disequilibrium (LD) and population recombination rate of *R. alpina* and *R. purpurea*: (**a**) LD decay of *R. alpina* and *R. purpurea*. The boxplot of LD decay is average *r*^2^ estimates with 1 kb bin; (**b**) population recombination rate boxplot of *R. alpina* and *R. purpurea* across 12 chromosomes.

**Figure 2 ijms-25-02265-f002:**
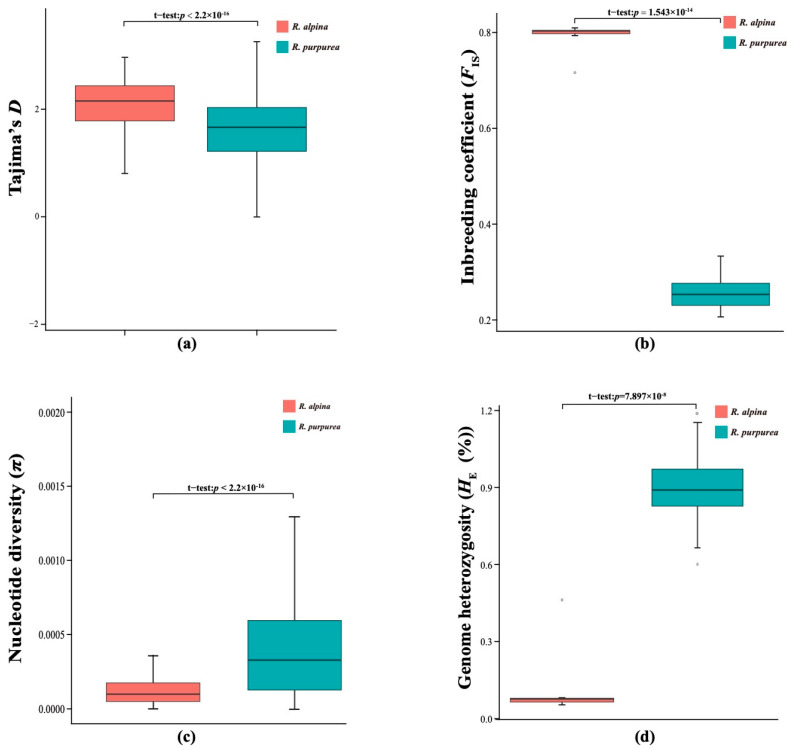
Genetic diversity parameters of *R. alpina* and *R. purpurea*: (**a**) Tajima’s *D* between two species; (**b**) *F*_IS_ between two species; (**c**) π between two species; (**d**) *H*_E_ between two species.

**Figure 3 ijms-25-02265-f003:**
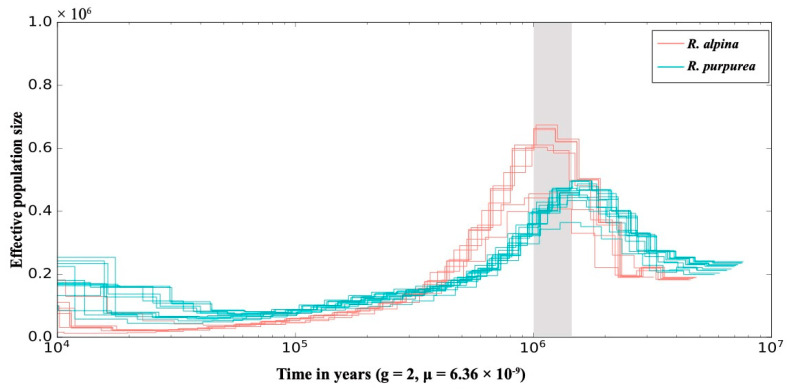
Demographic history of each individual inferred based on the pairwise sequential Markov coalescent (PSMC) model, colored by species (see legend). The X-axis shows the time in years, and the Y-axis shows the effective population size. Light-gray-colored shading marks the interval of significant decrease in effective population size at ~1.5–1.0 Ma. g, generation time; μ, mutation rate.

**Figure 4 ijms-25-02265-f004:**
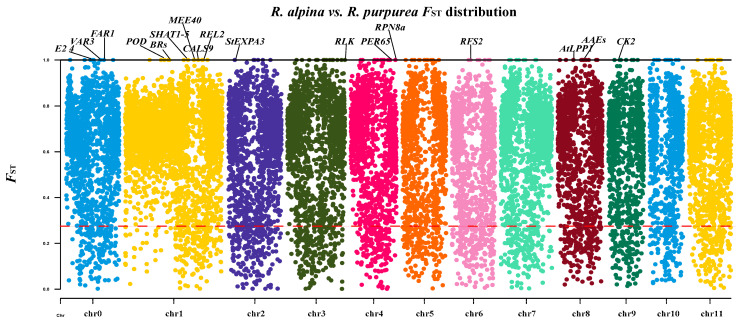
Manhattan plot of genome-wide *F*_ST_ between *R. alpina* and *R. purpurea* on each of the 12 chromosomes. The red dashed line indicates *F*_ST_ = 0.25 and the black solid line indicates *F*_ST_ = 1.

**Figure 5 ijms-25-02265-f005:**
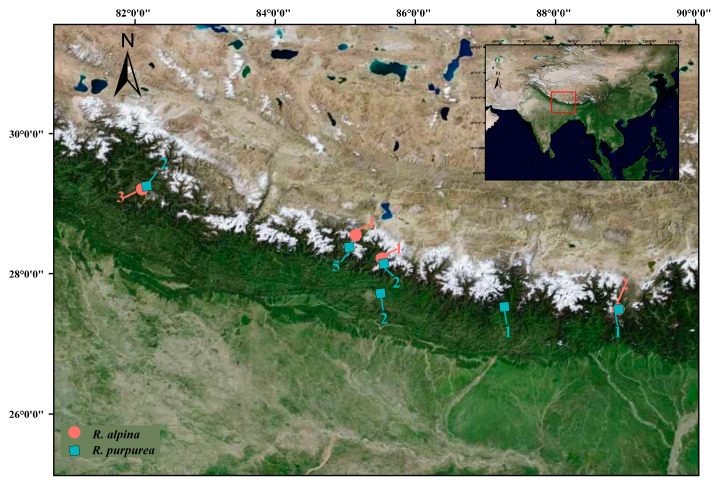
Sampling sites of *Roscoea alpina* and *R. purpurea.* The numbers beside the dots are the sample sizes of the sampling sites.

**Table 1 ijms-25-02265-t001:** Completely divergent (*F*_ST_ = 1) genes and function list for the comparison of *R. alpina* and *R. purpurea*.

Environmental Stresses	Genes	Annotations	References
Light intensity	*AAEs*	Participates in fatty acid and glycerolipid metabolism	[37]
Light intensity	*VAR3*	Part of a protein complex required for chlorophyll and carotenoid synthesis	[38]
Light intensity, biotic and abiotic stress	*SHAT1-5*	Provides high pod-shatter resistance	[39]
Light intensity, biotic and abiotic stress	*BRs*	Brassinosteroid biosynthesis	[40]
Biotic and abiotic stress	*REL2*	Controls leaf rolling	[41]
Biotic and abiotic stress	*E2*	Involved in plant biotic and abiotic stress responses	[42]
Biotic and abiotic stress	*CALS9*	Involved in sporophytic and gametophytic development	[43]
Biotic and abiotic stress	*StEXPA3*	Likely plays a role in tuber development	[44]
Biotic and abiotic stress	*RPN8a*	Determines leaf polarity	[45]
Biotic and abiotic stress	*MEE40*	May be involved in female gametophyte development	[46]
Circadian clock	*CK2*	Influences the circadian clock	[47]
Pathogen reduction	*RFS2*	Involved in the partial resistance to the spread of *Fusarium virguliforme* root infections	[48]
Pathogen reduction	*RLK*	Regulates plant development and defense responses	[49]
Pathogen reduction, hypobaric hypoxia	*PER65*	Involved in the response to wounding, pathogen attack, and oxidative stress	https://www.uniprot.org/uniprotkb/Q9FJR1/entry (accessed on 22 February 2023).
Hypobaric hypoxia, light intensity	*FAR1*	Modulates starch synthesis in response to light and sugar	[50]
Hypobaric hypoxia, low temperatures	*POD*	Carbohydrate or soluble sugar synthesis in plants of alpine regions	[51]
Molecular damage	*AtLPP1*	Reported to be induced by genotoxic stress (gamma ray or UV-B) and elicitor treatments with mastoparan and harpin	[52]

## Data Availability

The data and materials that support the findings of this study are available from the corresponding author upon reasonable request.

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
