# Peer review of "Genomic Insights into High-Altitude Adaptation: A Comparative Analysis of Roscoea alpina and R. purpurea in the Himalayas"

_ijms, 2024, doi:10.3390/ijms25042265_

Round 1

Reviewer 1 Report (New Reviewer)

Comments and Suggestions for Authors

1. The manuscript should include more in-depth literature review regarding to the local adaptation in high-altitudes.

2. The statistical significance tests between Tajima's D, genetic diversity and π are missing.

3. In Line 103-110, the Tajima's D, π  and other statistics are significantly different between R. purpurea and R. alpina. However, the boxplot of Tajima's D and π seemed not significantly different based on Figure 2. 

4. I can't agree with the conclusion in Line 108-109. The inapparent variation of Tajima's D and π  may due to the inadequate sampling size. 

5. Line 116-117, no statistical test to support the difference between the demographic history of two species.

6. The uniform distribution of Fst in Figure 4 may be a result of extremely small sample size.

7. Line 114, no valid support to select Fst=1 as candidate genes. Why not the 99.9% Fst outliers ?

8. Further analysis are required to validate the candidate genes under adaptation, such as SNP haplotype segregation analysis.

9. Are the SNPs within candidate genes synonymous ? No sufficient information provided.

10. Line 282-283, random sampling can reduce the sampling error, however can not solve the defect of small sample size.

11. Can the authors explain how to discriminate the locally adapted SNPs (Fst=1) from the confounding SNPs specific to either species but not related to local adaptation?

In summary, more in-depth analyses should performed to explore the genetic mechanism underlying the local adaption to high elevations.

Comments on the Quality of English Language

The writing logic and grammar can be further improved.

Author Response

Reviewer 2 Report (New Reviewer)

Comments and Suggestions for Authors

The paper will adequately structured, though the order of result presentation is a bit distorted where results are presented earlier than the methodology although it is known that this can be taken care of prior to publication. 

Comments on the Quality of English Language

The English used in the paper is convincing, but there is still a need to read it again to better fine-tune the article. 

Author Response

Reviewer 3 Report (New Reviewer)

Comments and Suggestions for Authors

Hill agriculture characterized by various types of features including hill topography that plays crucial role in species richness. Therefore, understanding the molecular mechanism of plant adaptation is essential to make the hill agriculture profitable. Here author showed the genetic adaptive mechanisms of high-altitude Himalayan endemic alpine genus Roscoea. Author reported total 58 completely divergent genes (FST = 1), of which only 17 genes were annotated to the function relates with the adaption to the high-altitude environments.   This study is novel and quite interesting and the science is strong. However, the following issues need to address before considering the manuscript.

1. In results line 80, author suggested to write the elaborative form of r-square (r2) instead of abbreviation. Or rewrite the sentence like “The average linkage disequilibrium (LD)….”

2. The sentence in line 133 is not clear. Author suggested to meaningful write-up by avoiding too many conjunctions.

3. If possible, author suggest to provide the qRT-PCR results of Candidate genes that associated with high-altitude adaptation.

In conclusion line 335-388 is irrelevant with the findings of the result, so author suggested delete the sentence or modified the sentence based on the obtained results.

Author Response

Reviewer 4 Report (New Reviewer)

Comments and Suggestions for Authors

In their manuscript the authors describe the resequencing of 7 and 13 individuals of two Roscoea species. By analysis of LD, Tajima‘s D, F-statistics, heterozygosity and nucletide diversity the authors try to track down genes which are involved in the adaptation to high altitude.

The approach of the authors sounds interesting and the statistics approaches used are also valid. However, overall the manuscript is difficult to read.

It starts of with a very short introduction that could well use a few more examples to adaptation to high altitudes in other plant species. However, this part is still easy to follow.

In the results section, though, I miss a little background information in each paragraph. In the paragraph on LD I miss a short introductory sentence saying „For getting an insight into …. LD was determined….

The paragraph on Tajima‘s D and the other measures is very, very short and so is the one on demographic history.

The section on candidate genes is short an not very precise. It seems that a true GO term analysis is missing. Indeed, the function of the genes they found seems to fit well with adaptation to high altitude, however, DNA repair, environmental stress etc. often show up in gene function.

The discussion is short and lacks clarity as well.

The description of material and methods is sufficient.

The conclusion section is in my view to definite. I think the authors identified possible candidate genes. They have no proof, not even by showing differential expression at different environmental conditions or something related. So saying „These completely divergent genes are likely the evolutionary innovations of R. alpina driven by harsh environmental stresses.“ is way beyond of what the authors can actually show.

Comments on the Quality of English Language

The language is fine, there are no major problems. Some parts of the results section do not read very fluent but maybe this is a matter of style.

Round 2

Reviewer 4 Report (New Reviewer)

Comments and Suggestions for Authors

The comments of the reviewers have been adequately addressed.

This manuscript is a resubmission of an earlier submission. The following is a list of the peer review reports and author responses from that submission.

Round 1

Reviewer 1 Report

Comments and Suggestions for Authors

The manuscript “Genomic insights into High-Altitudinal……and R. purpurea in the Himalayas” by Ya-Li Wang and coworkers provides a genome analysis of a few members of two Roscoea species, identifying particular highly divergent genes and various genome properties that differentiate the two species.  The work is adequately (if routinely) performed, and provides some interesting correlations, but the interpretations are often not appropriate, and further analysis is needed to make the correlations more compelling.

The major problem with the study is that it compares two very divergent species that have a non-overlapping range, and then concludes that the major differences between them are due to the differing adaptive selection pressures at their different habitat altitudes.  Of course, this is a likely possibility, but these are correlation studies, so there is zero justification to make these conclusions.  Most or all of the differences could be attributed to different founder effects on the two different species or because of their very different reproductive strategies.  High LD (which is basically the same thing as low recombination) and the various other effects they see on the genome are very routinely associated with a selfing strategy.  Also, very few samples (which the authors have for R. alpina) always lead to a conclusion of high LD.  Hence, I cannot encourage acceptance of this manuscript unless (a) the journal is satisfied with purely correlative studies and (b) the authors are required to tone down their conclusions to point out that they are dealing with correlations.

Some additional comments that should be addressed: 

(1)   The manuscript has a vast number of references, many of which are by the authors.  Of course, they want to increase their H-indices, but a smaller number of references would be better.  In this regard, the first paragraph states verities that are more than a century old, so this paragraph is not required.

(2)   On page 2, what does “deeply” diverged mean?  Of course they diverged, or they would not be separate species, but an approximate date of the divergence would be more useful than the “Miocene”, which covers 5 mya to 23 mya. 

(3)   Also on page 2, lines 82-83, the authors tell us that .55 is higher than .3, which I think most readers would already have figured out from lines 81-82. 

(4)   Also on line 83, the word “correlation” is not appropriate.

(5)   Lines 99-101 are not true.  It is homozygosity that increases LD, not stress.  Indeed, if more individuals die out because of stress, then there is a tendency toward inbreeding, but the direct cause of high LD has nothing to do with the stress.  If anything, genetic studies show stress can increase recombination rates.

(6)   On line 132, the word “response” is inappropriate.  The authors mean “outcome”.  Evolution itself is an outcome, not a response. 

(7)   Figure 2 is ridiculous.  The conclusion of the Figure is that these are two different species.  We knew that at the start.  I have not seen anyone putting two different species into such analyses.  These analyses are meant to show differences within species germplasm.  STRUCTURE and PCA analyses would be useful to assay on one species at a time, especially to see if it correlates with the locations of collection in any way.

(8)   On lines 183-189, the terminology and process are somewhat obscure.  I’ve never heard of “highly genetically divergence”, but the phrasing might be somewhat more understandable if the word “divergent” is used to replace “divergence”.  In general, I wonder why the authors are interested in divergent “windows” instead of divergent genes.  I also wonder how 128 went to 89 went to 77 went to 21.  If these are not arbitrary choices, then they need to be better explained here as well as in the Materials and Methods.

(9)   Line 191, I do not believe that “few pathogens” is a kind of severe environmental stress (line 190).

(10)                   In Table 1, such correlations are only interesting if they are compared to total gene content.  For instance, two of their 21 genes are related to molecular damage, thus a bit under 10%.  What percentage of genes overall are related to molecular damage?  Is it 10%?  If so, then they have no enrichment for this trait.

(11)                   I must have missed something, but I do not know what light “density” means.  Is it light intensity?

(12)                   I also have trouble seeing how tuber development, gametophyte development or leaf polarity have anything to do with “Biotic and abiotic stress”.  Are there any genes that do not have something to do with “Biotic and abiotic stress”?

(13)                   In Table 2, a map of collection sites would be nice.

(14)                   Lines 287-304 have a lot of causality attributed to correlations, and although these speculations are quite obvious and quite old (that is, made by many previous authors), that does not mean you can pretend causality.  In the 299-304 lines, the authors talk about shared traits among high altitude plants.  Were all of these listed plants inbreeders, or did some outcross?  The “adaptive syndrome of high elevation” could, of course, have been proposed long before this work was done or this manuscript was written, because they are common sense adaptations, but the authors need to look at both commonalities and exceptions in a comprehensive manner before such a “syndrome” is seriously proposed. 

Comments on the Quality of English Language

None

Reviewer 2 Report

Comments and Suggestions for Authors

The article describes the comparison of populations of Roscoea alpina and R. purpurea from Himalaya, and is aimed at providing insights into high altitudinal adaptation. The topic is interesting but the paper suffers from two major concerns:
- the poor quality of English makes it difficult to understand the approaches chosen and the results obtained
- the "populations" that are compared correspond to 7 R. alpina and 13 R. purpurea specimens that were resequenced. It is not clear that such small sample sizes allow to raise conclusions about genetic diversity, demographic history, and environmentaly adaptive genes. 
As a comparison, references cited in the text compare much more individuals: 
ref 75: 377 accessions of
Prunus, ref 96: 201 genotypes of Elymus sibericus, ref 97 :  253  tea plant accessions.

These two concerns make the article unconvincing and the conclusions questionable.

Below are more detailed comments (no list of grammatical errors is provided since the whole text requires to be corrected):

Line 87 : "quite" different. Such a terminology should be avoided. If the difference is significant, replace with "significantly different". 

Line 123 : lower tha those of R. alpina -> R. purpurea

line 132: "Genetic bottleneck is an evolutionary response of species to severe environmental changes which would decrease genetic diversity". The sentence is incorrect: genetic bottelneck is a consequence of the decrease of genetic diversity but not a "response", as written in the next sentence: " Thus, lower genetic diversity of R alpina could be the genetic consequence of adaptive evolution to the higher elevation in the Himalayas".

Line 186-187: 77 genes were annotated in the NR database but only 21 are in table 1 and "21 were annotated to specific functions": what are the functions of the remaining 56 genes?

Line 204 : define convergent evolution

Comments on the Quality of English Language

The article is very poorly written and hard to understand. Extensive corrections of English language are needed.

Reviewer 3 Report

Comments and Suggestions for Authors

Dear authors, 

Wang et al. provides valuable insights into the genomic signature of altitudinal adaptation through a very well-designed comparative study of two Himalayas endemic alpine (R. alpina, and R. purpurea), with distinct, and non-overlapping altitudinal distribution. The Himalayas, with having elevation gradient of over 8,000 meters within a narrow latitude, is one of the hotspots for biodiversity and speciation research, and the choice of the R. alpina, and R. purpurea as predominantly selfing, and outcrossing species, respectively are providing a very well experimental set up. In addition, the R. alpina being diverged from the ancestor of R. purpurea in about middle Miocene (~10-16 million years ago) provide enough evolutionary time for genomic divergence and adaptation.

The results of this study show that the genome of R. alpina as a species inhabiting the higher elevation harbors less genetic variation, higher level of LD (lower rate of recombination), Tajima’s D, and FIS, demonstrating genomic signature of adaptation to the high elevation. In such harsh environment, organisms are facing challenges associated with intense ultraviolet radiation, low temperature, hypoxia, and less interaction with pathogen influencing the function of various genes, morphological and physiological traits to facilitate the adaptation and optimize the survival, and reproduction. In this study, the authors used various population genomics tools to examine the genomic changes associated with adaptive mechanism to high- altitudinal environments particularly within the Himalayas.

I value this research highly, and here provide few comments aiming to improve the manuscript.

A general comment: please either do not abbreviate at all or abbreviate the first time the term appears in the text, and then only use the abbreviation. For example LD, PCA are constantly abbreviated throughout the text, and full definition is used!!

Line 78: linkage disequilibrium and population recombination landscape

Line 81: please provide information on the statistical test, and p-value for the significant difference in LD coefficients reported,

Line 87: please provide information on LD decay and the genomic window over which the LD decay was calculated – instead of saying the values are “quite different” please provide numbers, and if they are statistically or non-statistically different.

Line 98: make a space between “.” And “in some”.

Line 100-101: how much of the lower recombination, and higher LD can be explained by natural selection, and how much by reproduction mechanism (selfing). In addition, population size, and diversity also affect both the recombination rate, and LD … small populations, less diversity, less recombination and slower decay in LD. All these factors should be mention, and taken into account rather than indicating “natural selection” as a sole factor, all these evolutionary forces influence the phenomena.

Line 104-106: please provide higher quality figure. The figure heading is LD decay … in addition remove the word “plot” from the description, and provide more explanatory information. In addition for recombination rate on the figure please provide matrix system (cm/Mb). For the visually purpose (just a suggestion – optional) you can use the picture of these plants in the side of the graph to make it more appealing. At the current state the figure one is not in a publishing quality and rather like a figure draft.

Line 108: please provide in the main text or supplementary that how you pruned and filtered the SNPs.

Line 136: in figure 2 the species colors are different than figure 1 making it rather confusing.

In figure 2a, please change the “membership coefficient” to “Ancestry proportion” which is more informative and easier to read.

Line 137: remove “barplot”

Line 138: based on …

Line 139-138: the information on PCA are long and not explanatory. Just enough to mention that is PCA on these species, used for population clustering using XXX SNPs.

Scatter plot and not diagram … in general no need to mention the type of plots.

Line 141: in figure D, please write the full name on the Y-axis, for example Heterozygosity (He).

Line 141-143: remove the types of plots – provide more clear information. Also the title of figure 3 is not clear – which figure exactly is related to demographic history? We have genetic diversity, heterozygocity, inbreeding and Tajima’sD all showing the distribution of “genetic variation” in the genome.

Line 145: 1.6-1.0Ma – make a space, and also mention “million years ago” for the first time in line 145, and not 148.

Line 150: repetition of abbreviation (million years ago). Please avoid, I stop mentioning it from here.

Line 155: in figure 4 the color of species has totally changed. Please pick one color per species and use it throughout the text. Also for the abbreviation “g”, and “u” provide generation time, and mutation rate. The information provided in figure capture are obvious, come up with something like:

The demographic history of each individual/sample colored by species (see legend). All results were scaled using a generation time of two years and a mutation rate of (xx changes per site per generation).

Also pay attention to space between numbers and letters.

Line 158-159: in the phrase “The difference of demographic history suggests that R. alpina and R. purpurea within the Himalayas showed different response to climate changes.” Is very strong and changes in demographic history can be the results of many other things affecting the effective pop size, and climate change is one of them, but not the only one, so pelase soften this conclusion.

Line 181: remove “fdxz”

Line 181: using only Fst to identify environmentally adaptive genes is not sufficient. I suggest using a combination of tests which are based on the rations of polymorphism to divergence withing genes, and among genomic windows, such as homogeneity and Hudson–Kreitman–Aguadé tests, and low nucleotide diversity and high divergence.

Line 197-200. I suppose there are two sentences below, please make a separation.

“Based on our observation, budding time and flowering time are different between R. alpina and R. purpurea, CK2 gene may regulate the circadian clock to cope with the unpredictable climate between higher and lower altitude in the Himalayas.”

…. between R. alpina and R. purpurea.  CK2 gene may regulate ….

Line 212: where are all the genes with FST=1?

Line 216-217: based on the information below, there is sample size issue. Have you performed any test for sample size biase, or normalise the sample size?

“The Roscoea samples, seven individuals of R. alpina and thirteen individuals of R. purpurea, were collected from wild populations of Himalayas.”

Line 219: please specify the minor modification.

Line 223: what does “clean” means here in clean sequencing.

Line 226: how come that the reference genome Roscoea schneideriana is unpublish? It seems there is even chromosomal information on it, but why remained unpublished?

Line 244: remove “This is a table. Tables should be placed in the main text near to the first time they are

cited.”, and instead provide information in the caption. What is the genome size? And how much coverage you have here.

Line 276: “assembly. Then”. There are too much spacing here.

Best Wishes,

Comments on the Quality of English Language

Moderate editing of English language required

Round 2

Reviewer 3 Report

Comments and Suggestions for Authors

Dear Authors,

Many thanks for revising the manuscript and openly considering all suggestions and comments. The manuscript has been improved immensely, and its is much more clear, and sound.

I wish you all the best, and looking forward to see this work published.

Best Wishes,